# Conditional Control of CRISPR/Cas9 Function by Chemically Modified Oligonucleotides

**DOI:** 10.3390/molecules30091956

**Published:** 2025-04-28

**Authors:** Liangliang Wang, Yan Liu, Hongjun Song, Xue Zhang, Yang Wang

**Affiliations:** 1School of Biological and Pharmaceutical Engineering, Lanzhou Jiaotong University, Lanzhou 730070, China; 2Key Laboratory of Bioinorganic and Synthetic Chemistry (Sun Yat-sen University), Ministry of Education, Guangzhou 510006, China; liuy598@mail2.sysu.edu.cn (Y.L.); songhj5@mail2.sysu.edu.cn (H.S.); 3Key Laboratory of Tropical Biological Resources of Ministry of Education and One Health Institute, School of Pharmaceutical Sciences, Hainan University, Haikou 570228, China

**Keywords:** CRISPR/Cas9, guide RNA (gRNA), chemical modification, conditional control, gene editing

## Abstract

The CRISPR (clustered regularly interspaced short palindromic repeats) system has emerged as a revolutionary gene-editing tool with immense potential in gene therapy, functional genomics, and beyond. However, achieving precise spatiotemporal control of gene editing in specific cells and tissues while effectively mitigating potential risks, such as off-target effects, remains a key challenge for its clinical translation. To overcome these limitations, researchers have developed innovative strategies based on chemical modifications of oligonucleotides to enhance the precision, efficiency, and controllability of CRISPR/Cas9-mediated gene editing. By introducing conditional responsive elements, such as photosensitive groups, small-molecule responsive units, and supramolecular structures, they have successfully achieved precise spatiotemporal and dose-dependent regulation of CRISPR/Cas9 function. This review provides a comprehensive overview of recent advancements in gRNA regulation strategies based on chemical modifications of oligonucleotides, discussing their applications in improving the efficiency, specificity, and controllability of CRISPR/Cas9 editing. We also highlight the challenges associated with the conditional control of gRNA and offer insights into future directions for the chemical regulation of gRNA to further advance CRISPR/Cas9 technology.

## 1. Introduction

Since its development as a genome-editing tool in 2012 [1,2], the CRISPR/Cas9 system has rapidly become a cornerstone of modern life sciences due to its high efficiency and versatility [3]. The system primarily consists of the Cas9 protein and guide RNA (gRNA), which can exist as a single-guide RNA (sgRNA) or a complex of CRISPR RNA (crRNA) and trans-activating crRNA (tracrRNA) [1,4]. The mechanism of CRISPR/Cas9 involves the gRNA binding to a complementary target DNA sequence, guiding the Cas9 protein to the specific genomic location where it induces a double-strand break (DSB) (Figure 1a,b) [1]. The CRISPR/Cas9 system enables researchers to execute precise genetic modifications, including targeted knockout, site-specific insertion, and error correction, thereby serving as a versatile platform for both elucidating fundamental gene function and engineering potential therapeutic solutions [5,6,7]. While CRISPR-Cas9 technology holds revolutionary promise for genome editing, its clinical application remains constrained by unresolved technical limitations, particularly the issue of off-target effects caused by uncontrolled activity and non-specific DNA cleavage [8]. In addition, partial homology between gRNA and non-target DNA sequences can lead to unintended genomic mutations, potentially disrupting genomic stability and triggering adverse effects, such as malignant cell transformation. Therefore, reducing off-target effects and improving the precision of gene editing are critical for advancing the clinical translation of CRISPR/Cas9 technology.

In recent years, researchers have actively explored various strategies to overcome these limitations [9,10,11,12,13,14,15,16]. Notably, conditional control methods based on chemical modifications of oligonucleotides have emerged as a particularly promising avenue in CRISPR/Cas9 engineering research [17,18,19,20,21]. Unlike traditional approaches that focus on modifying the Cas9 protein, these methods offer greater flexibility and designability [22,23,24]. By introducing multifunctional responsive elements, such as photosensitive groups, small-molecule responsive units, host–guest recognition motifs, and so on, researchers can achieve precise temporal, spatial, and dose-dependent control of CRISPR/Cas9 activity (Figure 1c). In 2021, Zhang et al. systematically discussed the research progress in chemically diverse engineered guide RNAs, including length optimization, chemical modification strategies, and their conditional control applications in gene editing and regulation [21]. Different from previous reviews, the current review provides a comprehensive update: we not only critically evaluate recent breakthroughs in condition-responsive regulatory strategies based on oligonucleotide chemical modifications but also introduce novel regulatory mechanisms, such as light-responsive conformational constraints [25,26], small-molecule-induced allostery [27,28], and supramolecular host–guest recognition [29,30].

Collectively, this review systematically summarizes recent advances in gRNA regulation strategies based on chemical modifications of oligonucleotides and their applications in enhancing the efficiency, specificity, and controllability of CRISPR/Cas9-mediated gene editing. Furthermore, we examine the current challenges and future prospects of these strategies, offering insights into their potential for advancing both basic research and clinical applications.

## 2. Strategies for gRNA Regulation Based on Chemical Modifications of Oligonucleotides

As a critical component of the CRISPR/Cas9 system, gRNA plays a decisive role in guiding the entire gene-editing process. The gRNA regulatory strategies based on chemical modifications of oligonucleotides have opened up broad prospects for achieving the precise control of CRISPR/Cas9 function. Currently, regulatory strategies based on chemical modifications mainly include light-controlled, small-molecule-responsive, supramolecular regulation, and protein- or oligonucleotide-mediated approaches. Each of these strategies has distinct advantages, and as research progresses, new achievements and applications continue to emerge.

### 2.1. Light-Controlled Strategies

Light, as a non-invasive external stimulus with a high spatiotemporal resolution, offers unique advantages for the precise regulation of the CRISPR/Cas9 system. Since Bhatia et al. first reported the application of photoactivatable oligonucleotides in the CRISPR system in 2016 [31], researchers have developed various light-controlled gRNA strategies by incorporating photosensitive groups, enabling precise spatiotemporal control of gene editing [32]. The mechanism of light-controlled gRNA relies on the introduction of photosensitive groups, which undergo conformational changes or photochemical reactions under specific wavelengths of light. These changes regulate either the binding affinity between gRNA and the Cas9 protein or the complementary pairing between gRNA and target DNA. Currently, predominant photosensitive groups employed in photoresponsive systems, including 6-nitropiperonyloxymethyl (NPOM) [33,34], 7-diethylaminocoumarin (DEACM) [25,35,36], azobenzene [37], and so on [38], have emerged as critical molecular components enabling the development of light-controlled CRISPR platforms. These light-responsive moieties demonstrate distinct photochemical properties that facilitate the precise spatiotemporal regulation of genome-editing activities through light-controlled CRISPR systems.

#### 2.1.1. Photocaging Strategies

Photocaging has emerged as a predominant strategy for achieving light-controlled CRISPR systems, utilizing photo-responsive moieties (termed photocages) that are strategically conjugated to key structural motifs of the guide RNA. These modifications target specific sites, including nucleobase residues, 2′-hydroxyl groups, and phosphate backbone linkages, creating molecular switches that enable the precise spatiotemporal regulation of CRISPR-Cas activity through light illumination.

Caging nucleobases: Caged nucleobases block Watson–Crick hydrogen bonding to render oligonucleotides inactive until photocleavage of the caging groups restores nucleic acid base pairing. In 2020, Deiters’ and Stevens’ groups independently developed light-activated strategies, employing NPOM-caged nucleobases to disrupt gRNA–target DNA hybridization [39]. Deiters incorporated NPOM-caged uridine and guanosine, evenly distributed within the spacer region of a single gRNA (Figure 2a) [40], while Stevens installed two NPOM-caged thymidines in place of canonical uridines within the crRNA spacer region (Figure 2b) [41]. The caged crRNA complexed with its cognate tracrRNA exhibited similar functionality to Deiters’ single-caged gRNA design, both of which could be readily activated via 365 or 405 nm irradiation for 5 min to achieve spatiotemporally controlled gene editing in zebrafish embryos. Also in 2020, Ha et al. developed a very fast CRISPR (vfCRISPR) system by selectively replacing two or three uracil bases of crRNA with NPOM-caged thymidine. The vfCRISPR system retained its ability to bind to target DNA but remained catalytically inactive due to steric hindrance, preventing full DNA unwinding and nuclease activation. Upon irradiation with 365 nm or 405 nm light, the system achieved a submicron spatial resolution and a subsecond temporal resolution in living cells, providing an exceptionally precise tool for the spatiotemporal control of gene editing [42]. Moreover, Wang et al. proposed a strategy that uses short single-stranded DNA with photocleavable modification as invading oligonucleotides to switch on/off the activity of sequence-engineered gRNA by photo-controlled strand displacement reactions [33].

Caging 2′-OH: The structural characterization of the Cas9/gRNA ribonucleoprotein complex, combined with structure-guided chemical modification studies of the sgRNA scaffold, demonstrated that 2′-hydroxyl groups in the guide RNA’s ribose sugars mediate critical interactions with conserved amino acid residues within the Cas9 endonuclease [4,10]. This finding suggests the potential utility of incorporating photosensitive moieties at the 2′-OH positions of gRNA as a strategic approach to enable spatiotemporal control over Cas9 complex activity. Kool et al. developed a universal method for post-synthetic modification of RNA at the 2′-OH position using 1,1′-carbonyldiimidazole (CDI)-mediated activation [43,44]. Leveraging this chemical platform, Zhou et al. strategically modified the 2′-OH groups of in vitro-transcribed guide RNA (gRNA) using multiple ortho-nitrobenzyl photolabile protecting groups (DMNEC). This caging strategy effectively rendered the gRNA biologically inert until precise photoactivation through ultraviolet light exposure, enabling the spatiotemporal control of gene-editing activity in biological systems (Figure 2c) [45]. Subsequently, they extended their methodology to engineer a photoactivatable CRISPR sgRNA-based regulatory system by integrating vinyl ether (VE) group-mediated bio-orthogonal reactions with 9,10-phenanthrenequinone (PQ) photochemistry under visible-light irradiation (Figure 2d) [46]. The described reaction mediates the site-specific conjugation of sterically bulky chemical moieties to single-guide RNA (sgRNA), effectively perturbing its tertiary structure through steric hindrance effects. This structural modulation enables the precise spatial–temporal inactivation of CRISPR-Cas9 ribonucleoprotein complexes, achieving the conditional termination of gene-editing activity while significantly attenuating off-target effects. However, the presence of multiple 2′-OH groups in gRNA creates inherent challenges in elucidating the exact number of modification sites. More critically, extensive 2′-OH caging necessitates prolonged light activation duration, which could raise potential cytotoxicity concerns due to extended exposure to photoirradiation. To mitigate over-modification issues, a site-specific photoactivatable gRNA system was developed by selectively modifying key 2′-OH groups in the gRNA seed sequence that mediate critical interactions with the Cas9 protein, which is of great practical value for achieving rapid and biologically compatible temporal control (Figure 2e) [47].

Phosphate backbone modification: Xiang et al. developed a general method for site-specific derivations based on 2′-O-methyl ribonucleotide phosphorothioate (PS-2′-OMe) [48]. By taking advantage of the efficient and specific reaction between PS-2′-OMe and aryl methyl bromides (Aryl-Br), they precisely attached the light-sensitive group (coumarin amine (CM)) to the phosphodiester backbone of gRNA, enabling the controlled release of caged gRNA under specific stimuli (Figure 2f). To verify the universality of this strategy, the research team successfully achieved visible-light-triggered and reactive oxygen species (ROS)-responsive CRISPR-Cas9-mediated gene editing in human cells. This strategy bypasses the direct modification process of solid-phase synthesis, providing a universal platform for the site-specific functionalization of gRNA and allowing the flexible introduction of various functional chemical groups.

#### 2.1.2. Photocleavage Strategies

Photocleavage is a strategy for the light-controlled manipulation of nucleic acids using photocleavable linkers. One of the widely used early methods involves incorporating photolabile moieties into the oligonucleotide backbone. Upon irradiation, the oligomer is cleaved, causing the modified nucleic acid to fragment and lose its functionality [3,49,50]. Alternatively, nucleic acids can be modified to form stable secondary structures, such as hairpin or cyclic structures. Upon light exposure, these structures undergo cleavage, releasing the nucleic acid strand, which can then target the desired gene [51].

In 2016, Bhatia et al. pioneered a light-controlled CRISPR/Cas9 system employing photocleavable single-stranded DNA as molecular shields. The approach involved designing light-responsive DNA protectors complementary to the spacer region of gRNA, which hinder the hybridization between the CRISPR complex and target double-stranded DNA until UV irradiation triggers protector dissociation (Figure 3a) [31]. However, compared with other photochemical-controlled CRISPR/Cas9 systems, the functional control of the CRISPR/Cas9 complex by DNA inhibitors was not very significant in the cellular context. Ha et al. designed photocleavable guide RNAs (pcRNAs) that endow the CRISPR/Cas9 system with a built-in mechanism for light-based deactivation. pcRNA enables the rapid and complete inactivation of Cas9 within one minute of light exposure, thereby enhancing the specificity of wild-type CRISPR/Cas9 systems through the temporal control of gene-editing activity (Figure 3b) [52]. In contrast with conventional photo-controlled CRISPR/Cas9 systems that rely on the photocleavage of gRNA components, Tang and colleagues developed an innovative light-inducible strategy for CRISPR/Cas9 regulation through the 5′-terminal conjugation of vitamin E via a photolabile linker. This novel approach achieved temporary system inactivation through steric hindrance from the vitamin E moiety, with subsequent light irradiation triggering linker cleavage to restore CRISPR functionality [53]. This method innovatively regulates system activity by loading external functional modules, rather than relying on traditional interference with nucleic acid pairing or recognition mechanisms.

Chemical crosslinking can rapidly identify RNA–protein and RNA–nucleic acid molecular interactions, both inter- and intra-molecularly. Devaraj et al. developed an RNA-CLAMP method, which allows one to clamp two stem-loops in the single-guide RNA (sgRNA) of the CRISPR/Cas9 gene-editing system with a photocleavable linker, completely inhibiting gene editing. Visible-light irradiation cleaved the photocleavable linker and restored gene editing with a high spatiotemporal resolution (Figure 3c) [25]. By designing two photocleavable linkers (DEACM and NB) that respond to different wavelengths of light and using guide RNAs with different guide sequences, they achieved multiplexed photoactivation of gene editing in mammalian cells. Similar to the site-specific enzymatic crosslinking of the stem-loops of sgRNA, the Cheng group developed another novel strategy based on intramolecular cyclization for conformational restriction, enabling a light-controlled CRISPR/Cas9 editing system with covalently cyclized gRNA involving photosensitive linkers (Figure 3d) [26]. The study established a photoactivatable guide RNA-based strategy that enables the spatiotemporally controlled CRISPR/Cas9-mediated editing of the MSTN gene in zygotes, thereby achieving the precise regulation of cellular developmental trajectories and tissue-specific differentiation programs. Recently, a wavelength-selective orthogonal photoactivation strategy was developed through the dual functionalization of gRNA with photocleavable linkers (NPOM and DEACM) exhibiting differential spectral responsivity (405 nm vs. 450 nm) (Figure 3e) [35]. This multiplex manipulation provided an effective pathway for the precise temporal control and fully orthogonal control of CRISPR/Cas9 function by treatments with two different wavelengths of light irradiation in the cellular context.

Although light-controlled gRNA strategies offer the advantages of a high spatiotemporal resolution and non-invasiveness, ultraviolet light may cause phototoxicity to cells, and its limited penetration depth makes it challenging to achieve gene editing in deep tissues. In the future, it will be necessary to develop more efficient, low-toxicity photosensitive groups and explore novel light sources (such as near-infrared light) to enhance their potential for in vivo applications.

### 2.2. Small-Molecule-Responsive Strategies

Small-molecule-responsive strategies offer another feasible pathway for the conditional control of the CRISPR/Cas9 system [54,55,56,57]. Compared with light-based control methods, small molecules have limitations in localized activation and require the consideration of factors such as cell toxicity, membrane permeability, and stability during application. However, small molecules are easy to handle, do not require specialized equipment, and can penetrate deep or opaque tissues. Additionally, they are dose-responsive, allowing for the flexible adjustment of dosage to achieve varying degrees of regulation.

The Staudinger reduction reaction, which involves aryl azides and phosphine compounds, has been widely recognized for its high specificity, efficiency, and bio-orthogonality in biological systems. Recently, this reaction has been employed by Kool [58] and Zhou [59,60] et al. to acetylate and protect multiple ribose 2′-OH groups of gRNA, thereby inhibiting the activity of the CRISPR system. When activation of the CRISPR/Cas9 system is required, the addition of phosphine compounds (R_3_P) triggers the Staudinger reaction, rapidly releasing the acylated protecting groups and restoring CRISPR/Cas9 activity (Figure 4a). Recent studies have shown that bis-nicotinyl azide (BIN) can achieve crosslinking reactions with the 2′-OH groups of different RNA strands [61]. This crosslinking strategy has been utilized to conditionally control the pairing of sgRNA with target DNA sequences, effectively inhibiting the system until the protective groups are removed by the phosphine reductant, thereby activating the system (Figure 4b) [27]. Wang et al. developed a method where ninhydrin reacts with guanine on the gRNA to inhibit CRISPR/Cas9 activity, and subsequently, the activity is restored by adding guanosine triphosphate (GTP) (Figure 4c) [62].

The inverse electron-demand Diels–Alder (inv-DA) reaction between norbornene (NB) and tetrazine (TZ) derivatives has been applied in biological systems for its catalyst-free nature, non-toxic byproducts, and excellent chemical tunability [63,64]. This NB-TZ conjugation chemistry enables the development of functional loss tools, providing a “negative” approach to study RNAs. Unlike phosphine-induced activation, the introduction of NB into the 2′-OH of gRNA leads to the functional loss of the CRISPR system through the polymerization of tetrazine (TZ) in reaction with NB-gRNA (Figure 4d) [28].

Leveraging the instability of boronic acid (BO) in the presence of reactive oxygen species like hydrogen peroxide (H_2_O_2_) [65], Zhou et al. developed a nucleobase borylation strategy using 4-(bromomethyl)phenylboronic acid (B1Br) to modify sgRNA, enabling it to respond to H_2_O_2_ [66]. This modification inhibited sgRNA-Cas9 binding, disrupting target DNA recognition. Upon reaction with H_2_O_2_, the boronic acid group in the masked sgRNA was converted to phenol, followed by 1,6-elimination and the release of the masking group, restoring sgRNA functionality and resulting in the complete switching of the CRISPR system between “off” and “on” states (Figure 4e). Additionally, nucleobase modifications with 5-formylcytosine (5fC) and 5-carboxylcytosine (ca5C) preserve gRNA functionality [67,68]. However, the conversion to dihydrouracil (DHU) using either a borane–pyridine complex (BPC) or 2-picoline–borane complex (2-PBC) inhibits Cas9 activity in a concentration-dependent manner, thereby preventing target DNA cleavage (Figure 4f). This chemical reaction-based strategy, which generates mutant regulation, has been effectively extended to studies involving Cas13a and the interactions of RNA aptamers such as Pepper and hydroxybenzyl cyanine (HBC).

Small-molecule regulation of gRNA function has been validated in mammalian cells and is highly attractive in animal model studies, as it can overcome the limitations of light control and provides strong support for in-depth research into gene function and disease treatment mechanisms.

### 2.3. Supramolecular Recognition Strategies

Supramolecular chemistry focuses on the formation of complex and ordered systems through non-covalent interactions between molecules, offering new perspectives for the regulation of functional nucleic acids [69,70]. Xiong et al. modified gRNA with adamantyl groups and used cucurbit[7]uril (CB7) to non-covalently bind to the adamantylated gRNA, regulating the interaction between RNA and proteins and, thus, achieving the regulation of CRISPR/Cas9 activity through supramolecular chemistry strategies (Figure 5a) [29]. The binding between biotin and streptavidin is one of the strongest non-covalent interactions known in nature. Due to the small molecular weight of biotin and the large steric hindrance of streptavidin, their interaction can effectively shut down the activity of CRISPR/Cas9, functioning as a “braking system” (Figure 5b) [30].

### 2.4. Other Condition-Responsive Strategies

#### 2.4.1. Protein- or Oligonucleotide-Mediated Regulation Strategies

Protein- or oligonucleotide-mediated gRNA regulation strategies leverage the specific interactions between proteins and gRNA, as well as oligonucleotides and gRNA, to achieve complex control over CRISPR/Cas9 functionality [71,72,73,74,75]. Wang et al. developed an enzyme-induced CRISPR (eiCRISPR) system, which consists of the Cas9 protein, a self-blocked inactive single-guide RNA (bsgRNA), and a chemically caged deoxyribozyme (DNAzyme) (Figure 6a). In the system, the DNAzyme is designed to remain inactive and “caged” until an activating enzyme, such as the NAD(P)H quinone oxidoreductase (NQO1), which is overexpressed in tumor cells, is introduced. Upon activation, the DNAzyme “uncages” and cleaves the bsgRNA, restoring its activity and enabling Cas9 to perform genome editing [76].

#### 2.4.2. Temperature-Responsive Strategies

Temperature can serve as a direct external stimulus for controlling nucleic acid caging, as introducing heat is relatively straightforward, and laboratory instruments for precise temperature control are common and cost-effective. Glyoxal, as a thermally responsive caging group, can react with nitrogen groups on the Watson–Crick–Franklin face of nucleobases to form stable bis-hemiaminal adducts [77]. These adducts can be reversed by heating in weakly alkaline conditions, enabling the application of thermally reversible glyoxal caging for the controllable inhibition and complete reactivation of the CRISPR-Cas9 system (Figure 6b) [78]. Glyoxal modifications are easy to install and effectively disrupt nucleic acid structure and function. They also allow tunable and reversible control over the activity of nucleic acid aptamers, RNA-cleaving 10–23 DNAzyme, and more, providing a versatile new tool for various potential synthetic biology and biotechnology applications.

## 3. Conclusions and Prospects

The CRISPR/Cas9 system, guided by gRNA, has revolutionized gene editing, offering immense potential for both basic research and clinical applications. By developing conditionally responsive gRNA regulation strategies based on chemical modifications of oligonucleotides, researchers have achieved precise spatiotemporal and dose-dependent control over CRISPR/Cas9 activity, significantly reducing off-target effects and enhancing the safety and specificity of gene editing. These advancements, including light-responsive, small-molecule, and supramolecular strategies, have opened new avenues for conditional control of CRISPR/Cas9, enabling a deeper exploration of gene function and more effective therapeutic interventions.

Despite these successes, several challenges remain. Current light-control strategies, while offering a high spatiotemporal resolution, are limited by the phototoxicity of UV light and poor tissue penetration [79]. Small-molecule regulation, though versatile, faces issues with biocompatibility and cellular toxicity. Supramolecular approaches, reliant on non-covalent interactions, struggle with stability and reversibility. Future research should focus on developing novel photosensitive groups responsive to near-infrared light [80,81], low-toxicity small molecules, and reversible supramolecular systems. Additionally, integrating multimodal regulation strategies, such as combining light, pH, and enzyme activation, could enable more precise and context-specific gene editing.

Looking ahead, the continued development of innovative conditional response strategies, such as ultrasound- or magnetic-field-responsive gRNA [82,83], along with advancements in site-specific chemical modifications, will further expand the potential of CRISPR/Cas9 technology. These efforts will pave the way for more sophisticated and targeted gene-editing applications, ultimately advancing precision medicine and the treatment of complex diseases. By tailoring gRNA modifications to respond to specific cellular or environmental cues, researchers can further enhance the precision and safety of CRISPR/Cas9-based therapies, bringing us closer to realizing the full potential of this transformative technology.

While this review primarily focused on the conditional control of CRISPR/Cas9 function through chemically modified oligonucleotides, the underlying principles of guide RNA regulation via chemical modifications are broadly applicable to other CRISPR-Cas systems [84], including Cas12 variants (such as Cas12a and Cas12b) and RNA-targeting Cas13 systems. For example, Chen et al. successfully developed a novel class of photocontrolled star-shaped multivalent crRNAs for the precise spatiotemporal control of CRISPR/Cas9 and Cas12a editing systems, demonstrating how these chemical modification strategies can be adapted across different CRISPR platforms [36].

## Figures and Tables

**Figure 1 molecules-30-01956-f001:**
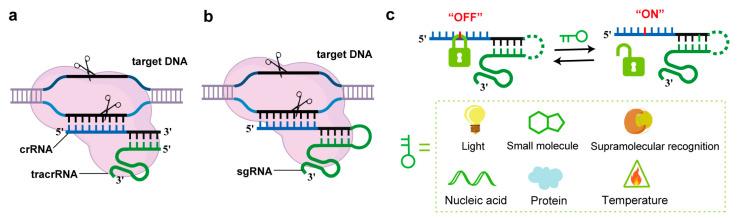
(**a**) Cas9 programmed by crRNA: tracrRNA; (**b**) Cas9 programmed by sgRNA; (**c**) schematic diagram of conditional control of CRISPR/Cas9 function by chemically modified oligonucleotides.

**Figure 2 molecules-30-01956-f002:**
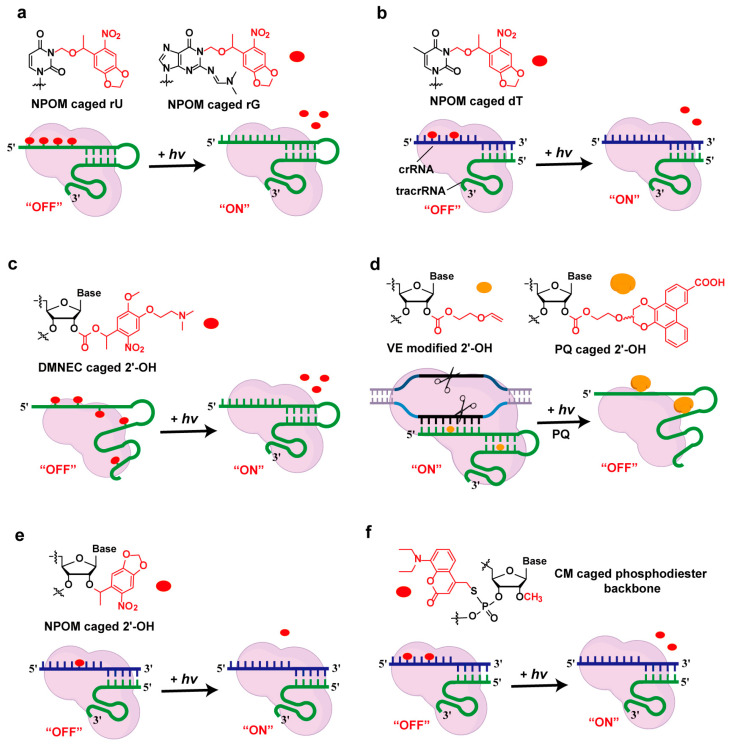
Photocaging strategies for controlling gRNA function. (**a**) NPOM-caged uridine and guanosine distributed in the spacer region of sgRNA; (**b**) NPOM-caged thymidine substitutions in the spacer region of crRNA; (**c**) DMNEC-caged 2′-OH groups on gRNA; (**d**) visible-light-catalyzed reaction of vinyl ether (VE) with 9,10-phenanthrenequinone (PQ) on sgRNA; (**e**) NPOM-caged 2′-OH group in the seed region of crRNA; (**f**) site-specific derivatization of gRNA using PS-2′-OMe and aryl bromide (Aryl-Br) for light-triggered release.

**Figure 3 molecules-30-01956-f003:**
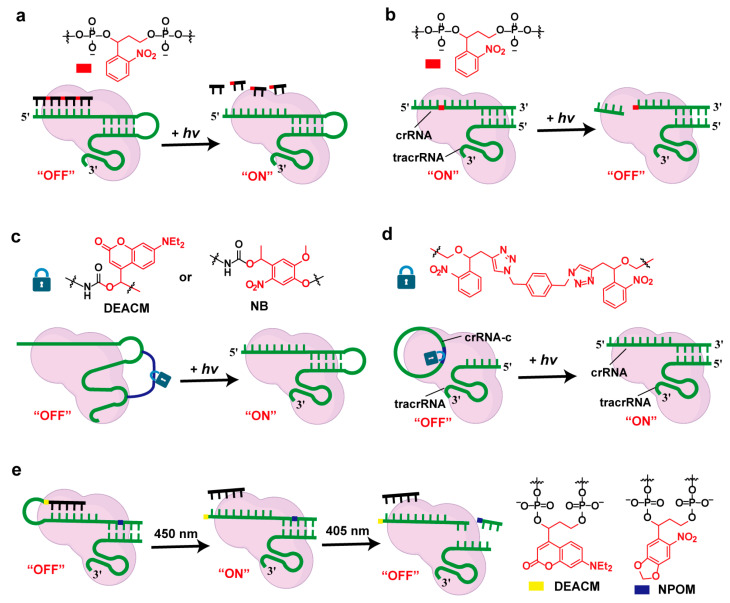
Photocleavage strategies for controlling gRNA function. (**a**) Light-cleavable single-stranded DNA “protector” blocking gRNA–target DNA hybridization; (**b**) photocleavable guide RNA (pcRNA) for rapid inactivation of Cas9; (**c**) photoactivation of site-specific enzymatically crosslinked sgRNA; (**d**) photoactivation of conformationally constrained circular gRNA with photocleavable linkers; (**e**) orthogonal dual-wavelength photoregulation of gRNA using NPOM and DEACM photocleavable linkers.

**Figure 4 molecules-30-01956-f004:**
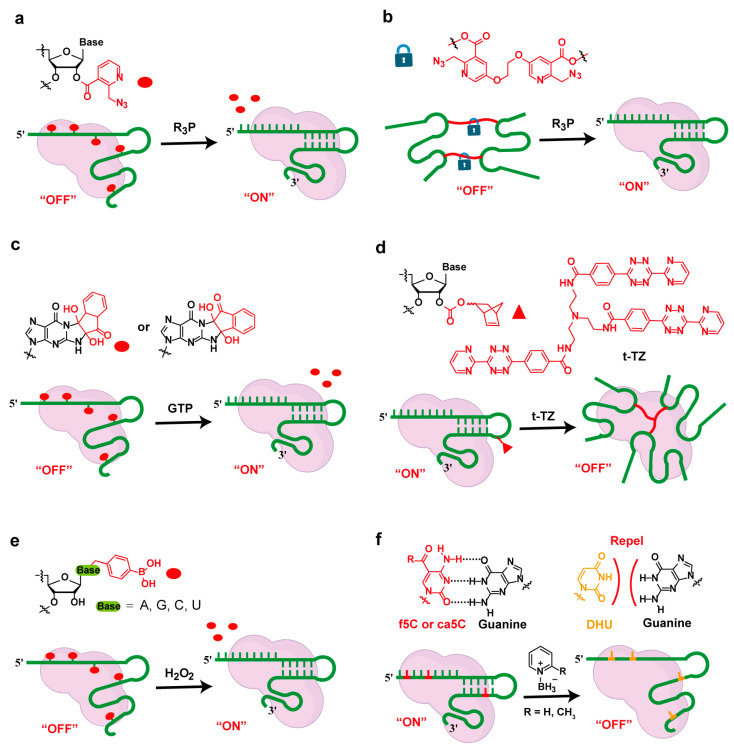
Small-molecule-responsive strategies for controlling gRNA function. (**a**) Staudinger reduction reaction for deprotection of acylated gRNA; (**b**) azide reduction reaction for activation of sgRNA; (**c**) ninhydrin-mediated inhibition and GTP-induced activation of CRISPR/Cas9; (**d**) norbornene–tetrazine ligation for CRISPR system inactivation; (**e**) hydrogen peroxide-responsive nucleobase modification for sgRNA activation; (**f**) chemical conversion of 5-formylcytosine (5fC) or 5-carboxylcytosine (ca5C) to dihydrouracil (DHU) for CRISPR/Cas9 inhibition.

**Figure 5 molecules-30-01956-f005:**
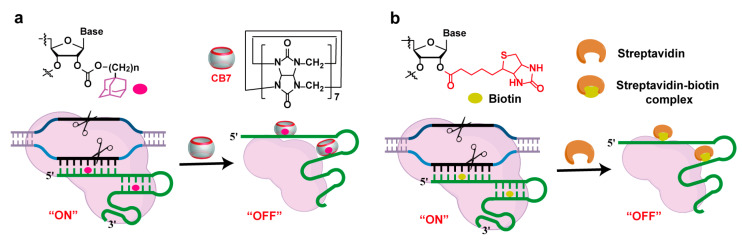
Supramolecular recognition strategies for controlling gRNA function. (**a**) CB7–adamantane complex for regulating RNA–protein interactions; (**b**) biotin–streptavidin interaction for CRISPR/Cas9 inactivation.

**Figure 6 molecules-30-01956-f006:**
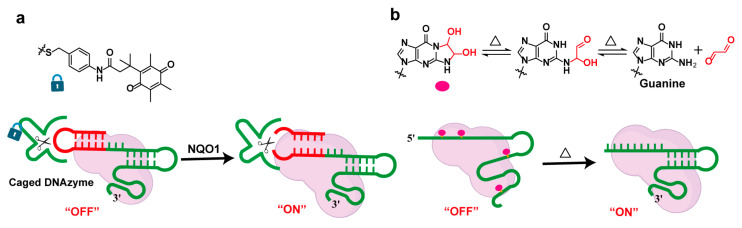
Other condition-responsive strategies for controlling gRNA function. (**a**) Enzyme-inducible CRISPR (eiCRISPR) system with self-blocking guide RNA (bsgRNA) and DNAzyme; (**b**) thermo-reversible glyoxal caging for CRISPR/Cas9 control.

## Data Availability

No new data were created or analyzed in this study.

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
