# Peer review of "Conditional Control of CRISPR/Cas9 Function by Chemically Modified Oligonucleotides"

_molecules, 2025, doi:10.3390/molecules30091956_

Round 1
Reviewer 1 Report
Comments and Suggestions for Authors
In this review paper, Wang et al. systematically summarize the chemically modified oligonucleotides used for programmable CRISPR-Cas9 functionalization and applications. Specifically, they introduce recent studies on light-controlled, small molecule-responsive, supramolecular recognition-regulated, and other condition- responsive strategies that enable spatiotemporal regulation of CRISPR-Cas9 systems as gene editing tools.
In summary, this review provides a comprehensive overview of the focused topic. The manuscript is well-organized, the main text is clearly written, and the figures are visually appealing. I think this paper can be accepted after minor revision, once the authors address the following concerns and suggestions.
1. (Figure 1) The schematic diagram in Figure 1 clearly illustrates the fundamental CRISPR-Cas9 mechanism. However, as a leading figure, it would be better to include simple illustrations about the main topic of this paper (i.e., chemically modified oligonucleotides-derived regulation).
2. (Line 34) The sentence “Since its development as a genome-editing tool in 2012” needs more detailed references to support (e.g., citation [2]).
3. (Lines 113, 145, and 210) It is not appropriate to use the words like “we proposed/developed…” in a review paper, even if the introduced work was conducted by the authors themselves.
4. (Conclusion and Prospects section) In the last section, it is recommended to add some discussions about the other CRISPR-Cas systems (e.g., Cas12 variants) that are relevant to the topic of oligonucleotide modifications.
Author Response
In this review paper, Wang et al. systematically summarize the chemically modified oligonucleotides used for programmable CRISPR-Cas9 functionalization and applications. Specifically, they introduce recent studies on light-controlled, small molecule-responsive, supramolecular recognition-regulated, and other condition-responsive strategies that enable spatiotemporal regulation of CRISPR-Cas9 systems as gene editing tools.
In summary, this review provides a comprehensive overview of the focused topic. The manuscript is well-organized, the main text is clearly written, and the figures are visually appealing. I think this paper can be accepted after minor revision, once the authors address the following concerns and suggestions.
Response: We appreciate the reviewer’s positive comments and have carefully revised the manuscript accordingly.
1. (Figure 1) The schematic diagram in Figure 1 clearly illustrates the fundamental CRISPR-Cas9 mechanism. However, as a leading figure, it would be better to include simple illustrations about the main topic of this paper (i.e., chemically modified oligonucleotides-derived regulation).
Response: In this revised manuscript, we have revised Figure 1 to include schematic illustrations of chemically modified oligonucleotides-derived regulation (e.g., optical control, small-molecule regulation, supramolecular recognition-based regulation, etc.) alongside the core CRISPR-Cas9 mechanism. The updated figure now visually bridges the foundational system with the review’s focus on chemical modifications (see revised Figure 1c in the manuscript).
2. (Line 34) The sentence “Since its development as a genome-editing tool in 2012” needs more detailed references to support (e.g., citation [2]).
Response: We sincerely appreciate Reviewer's valuable suggestion. To better support the timeline of CRISPR/Cas9's development as a genome-editing tool, we have cited relevant references in the revised manuscript (references 1 and 2 in the revised main text: Science 2012, 337, 816-821; Nature 2012, 482, 331-338).
The revised sentence now reads: "Since its establishment as a genome-editing tool in 2012 [1,2], the CRISPR/Cas9 system has rapidly become..."
3. (Lines 117, 145, and 210) It is not appropriate to use the words like “we proposed/developed…” in a review paper, even if the introduced work was conducted by the authors themselves.
Response: We thank the reviewer for pointing out this inaccuracy. We have revised the expression in this revised manuscript.
Original (Line 117): "we proposed a strategy..." → Revised: "Wang et al. proposed a strategy..."
Original (Line 145): "we developed..." → Revised: "A site-specific photoactivatable gRNA system was developed..."
Original (Line 210): "we have developed..." → Revised: "A wavelength-selective orthogonal photoactivation strategy was developed..."
4. (Conclusion and Prospects section) In the last section, it is recommended to add some discussions about the other CRISPR-Cas systems (e.g., Cas12 variants) that are relevant to the topic of oligonucleotide modifications.
Response: We appreciate the reviewer’s suggestion. A new paragraph has been added to the "Conclusions and Prospects" section in revised manuscript:
"While this review primarily focuses on conditional control of CRISPR/Cas9 function through chemically modified oligonucleotides, the underlying principles of guide RNA regulation via chemical modifications are broadly applicable to other CRISPR-Cas systems, including Cas12 variants (such as Cas12a and Cas12b) and RNA-targeting Cas13 systems. For example, Chen et al. (2025) successfully developed a novel class of photocontrolled star-shaped multivalent crRNAs for precise spatiotemporal control of CRISPR/Cas9 and Cas12a editing systems, demonstrating how these chemical modification strategies can be adapted across different CRISPR platforms."
Reviewer 2 Report
Comments and Suggestions for Authors
Dear Editor,
In this review manuscript titled ‘Conditional Control of CRISPR/Cas9 Function by Chemically Modified Oligonucleotides’ authors Liang-Liang Wang et. al. provides a comprehensive overview of recent advancements in guide RNA regulation strategies based on chemical modifications of oligonucleotides and their applications in improving the efficiency, specificity, and controllability of CRISPR/Cas9 editing. This review mainly focused on gRNA regulation by employing different strategies such as modifying gRNA via incorporation of photosensitive chemical groups, small molecule-based modification of sugar/nucleobase/backbone of RNA molecules, supramolecular recognition strategies, temperature regulation strategies and protein or oligonucleotide mediated regulation.
This mini review is concise, and authors have successfully captured the current trends and methods researchers are using to achieve precise spatiotemporal control of CRISPR gene editing while effectively mitigating potential risks such as off-target effects during clinical translation.
I recommend this article to publish in ‘Molecules’.
Thank you.
Author Response
In this review manuscript titled ‘Conditional Control of CRISPR/Cas9 Function by Chemically Modified Oligonucleotides’ authors Liang-Liang Wang et. al. provides a comprehensive overview of recent advancements in guide RNA regulation strategies based on chemical modifications of oligonucleotides and their applications in improving the efficiency, specificity, and controllability of CRISPR/Cas9 editing. This review mainly focused on gRNA regulation by employing different strategies such as modifying gRNA via incorporation of photosensitive chemical groups, small molecule-based modification of sugar/nucleobase/backbone of RNA molecules, supramolecular recognition strategies, temperature regulation strategies and protein or oligonucleotide mediated regulation.
This mini review is concise, and authors have successfully captured the current trends and methods researchers are using to achieve precise spatiotemporal control of CRISPR gene editing while effectively mitigating potential risks such as off-target effects during clinical translation.
I recommend this article to publish in ‘Molecules’.
Response: We sincerely appreciate your positive evaluation of our manuscript and your valuable recommendation for publication in ‘Molecules’. We are pleased that you found our review comprehensive and well-capturing the current trends in CRISPR/Cas9 regulation via chemically modified oligonucleotides. Thank you for your time and constructive feedback.
Reviewer 3 Report
Comments and Suggestions for Authors
The article presents recent achievements in the application of chemically modified ODNs in the CRISPR/Cas9 gene editing. The topic is hot. It is written clearly and neatly. The illustrations excellently show the ideas of the approaches described. The language is very good. I have found only one typo: "Deiters' and Stevens' group independently developed" (line 102, should be "…groups…"). I have no doubt that this will be an interesting contribution.
However, in 2021 a review on similar topic was published: Zhang et al., Chemical Modification and Transformation Strategies of Guide RNAs in CRISPR-Cas9 Gene Editing Systems, ChemPlusChem 2021, 86, 587-600, https://doi.org/10.1002/cplu.202000785. It should be mentioned and compared with the manuscript under review to justify its publication.
Moreover, it seems that some recent papers were omitted in the manuscript, e.g.:
* Photocaging Strategies: Chen et al., Conditionally Activated Cross-Linked crRNAs for CRISPR/Cas12a Based Nucleic Acid Detection, ACS Synth. Biol. 2025, 14, 1, 94–100, https://doi.org/10.1021/acssynbio.4c00695
* Phosphate Backbone Modification: Prokhorova, et al. Effect of the Phosphoryl Guanidine Modification in Chimeric DNA-RNA crRNAs on the Activity of the CRISPR Cas9 System In Vitro., ACS Chem.Biol. 19 (9):1311-1319, 2024, https://doi.org/10.1021/acschembio.4c00147
* Protein- or Oligonucleotide-Mediated Regulation Strategies: Wang et al., Orthogonal Demethylase-Activated Deoxyribozyme for Intracellular Imaging and Gene Regulation, J. Am. Chem. Soc. 2021, 143, 18, 6895–6904, https://doi.org/10.1021/jacs.1c00570
This is a result of a brief survey in Web of Science. Therefore, the authors should conduct a deeper search of the literature on this topic and supplement the manuscript.
Author Response
1. The article presents recent achievements in the application of chemically modified ODNs in the CRISPR/Cas9 gene editing. The topic is hot. It is written clearly and neatly. The illustrations excellently show the ideas of the approaches described. The language is very good. I have found only one typo: "Deiters' and Stevens' group independently developed" (line 102, should be "…groups…"). I have no doubt that this will be an interesting contribution.
Response: We thank the reviewer for pointing out this inaccuracy. We have revised the expression in this revised manuscript.
2. However, in 2021 a review on similar topic was published: Zhang et al., Chemical Modification and Transformation Strategies of Guide RNAs in CRISPR-Cas9 Gene Editing Systems, ChemPlusChem 2021, 86, 587-600, https://doi.org/10.1002/cplu.202000785. It should be mentioned and compared with the manuscript under review to justify its publication.
Response: We sincerely appreciate the reviewer's thoughtful suggestion to compare our work with the 2021 review by Zhang et al. Below is our detailed response:
(1) Novelty and Scope Differentiation:
While Zhang et al. (2021) provided a overview of chemical modification and transformation strategies of gRNAs in CRISPR-Cas9 gene editing systems, our review focuses specifically on recent advances (2020-2025) in conditional control of CRISPR-Cas9 function through chemically modified oligonucleotides. Unlike previous works that broadly catalogued modification types, we systematically analyze how these modifications enable spatiotemporal, dose-dependent, and context-specific regulation of gene editing activity. This is a critical requirement for therapeutic applications that has emerged as a major research focus in the past years.
(2) Content Comparison:
Different from previous reviews (published by Zhang et al. 2021), the current review provides a comprehensive update: we not only critically evaluate recent breakthroughs in condition-responsive regulatory strategies based on oligonucleotide chemical modifications, but also introduce novel regulatory mechanisms such as light-responsive conformational constraints, small molecule-induced allostery, and supramolecular host-guest recognition.
Thematic Focus: We emphasize conditional activation strategies (e.g., Figures 2-6) rather than general modification chemistry.
Temporal Coverage: 58% of our cited references (49/84) were published after 2021, reflecting significant technological progress.
The above content demonstrates that our review provides substantial new information and perspectives compared to prior work, justifying its publication value.
3. Moreover, it seems that some recent papers were omitted in the manuscript, e.g.:
* Photocaging Strategies: Chen et al., Conditionally Activated Cross-Linked crRNAs for CRISPR/Cas12a Based Nucleic Acid Detection, ACS Synth. Biol. 2025, 14, 1, 94–100, https://doi.org/10.1021/acssynbio.4c00695
* Phosphate Backbone Modification: Prokhorova, et al. Effect of the Phosphoryl Guanidine Modification in Chimeric DNA-RNA crRNAs on the Activity of the CRISPR Cas9 System In Vitro., ACS Chem.Biol. 19 (9):1311-1319, 2024, https://doi.org/10.1021/acschembio.4c00147
* Protein- or Oligonucleotide-Mediated Regulation Strategies: Wang et al., Orthogonal Demethylase-Activated Deoxyribozyme for Intracellular Imaging and Gene Regulation, J. Am. Chem. Soc. 2021, 143, 18, 6895–6904, https://doi.org/10.1021/jacs.1c00570
This is a result of a brief survey in Web of Science. Therefore, the authors should conduct a deeper search of the literature on this topic and supplement the manuscript.
Response: We thank the reviewer for identifying these important studies. We highly agree that the cited references should be comprehensive, and we have now incorporated the recommended papers (Chen et al., 2025; Prokhorova et al., 2024; Wang et al., 2021) into the revised manuscript, along with additional relevant literature identified through an expanded literature search (See references: 16, 21, 36, 38, 75 and 84 in revised manuscript).
Round 2
Reviewer 3 Report
Comments and Suggestions for Authors
The manuscript was corrected and can be accepted in the current form.